# Sustainability and the Thoroughbred Breeding and Racing Industries: An Enhanced One Welfare Perspective

**DOI:** 10.3390/ani13030490

**Published:** 2023-01-31

**Authors:** Lorann Stallones, Phil McManus, Paul McGreevy

**Affiliations:** 1Department of Psychology, College of Natural Sciences, One Health Institute, Colorado State University, Fort Collins, CO 80523-1612, USA; 2School of Geosciences, Faculty of Science, The University of Sydney, Sydney, NSW 2006, Australia; 3One Welfare Research Institute, Faculty of Science, Agriculture, Business and Law, University of New England, Armidale, NSW 2350, Australia

**Keywords:** horseracing, welfare, safety, climate change, social license to operate

## Abstract

**Simple Summary:**

This article addresses the problems that face Thoroughbred breeding and racing globally as the sector seeks to become sustainable. It considers the benefits and deleterious impacts of the sector on three broad groups of stakeholders: horses, humans within the sector, and the physical environment. The authors propose an “Enhanced One Welfare Framework” to guide discussions that revisit the way horses are used for sport and moderate racing’s effects on humans and the planet. Using the Five Domains model that has become established as a framework through which to assess animal welfare, we identify the effects of the physical impacts of horse management and training on the mental state of horses. In a novel step, we apply the same model to consider the sector’s impact on its personnel. In contrast to sustainability approaches that focus primarily on animals and call to ban horseracing, we explore pathways that could allow the beneficiaries and sponsors of racing to encourage incremental improvements in practice across the sector.

**Abstract:**

As society debates the use of animals in sport, entertainment, and leisure, there is an increasing focus on the welfare, social, and ecological impacts of such activities on the animals, human participants, people close to them, and the physical environment. This article introduces the “Enhanced One Welfare Framework” to reveal significant costs and benefits associated with Thoroughbred breeding and racing globally. In addition, relative to calls to ban horseracing and similar activities as part of sustainability approaches that focus chiefly on animals, the “Enhanced One Welfare Framework” is better positioned politically to guide discussions that renegotiate the conditions under which horses are used for sport and the impact racing has on humans and the planet. In 2020, the International Federation of Horseracing Authorities issued its minimum horse welfare standards based on the Five Domains model, positioning lifelong horse welfare as “fundamentally important to the viability and sustainability of the industry”. In this article, we critique the One Welfare framework’s historic lack of focus on sport and enhance it by including sport, leisure, and entertainment and framing it within the Five Domains model. We offer a novel extension of the Five Domains model beyond animal welfare to consider human welfare and the physical environmental impacts of the sport, leisure, and entertainment industries and propose innovations that may help thoroughbred breeding and racing assure a sustainable future.

## 1. Introduction

Sustainable development, as proposed by the World Commission on Environment and Development (WCED) and through the United Nations’ Sustainable Development Goals (SDGs), does not consider animals appropriately. Non-human animals appear in key texts, such as Our Common Future [1], primarily as resources for human use [2,3,4,5,6,7]. Within this framework, three forms of animals were identified as: ‘wildlife implicitly being a constituent of ecosystems and biodiversity, pests to be controlled, and “livestock”’ [5]. This construction of animals as wildlife, pests, or food sources overlooks the importance of animals in sport.

The debates about the use of animals in sport, entertainment, and leisure are growing, particularly in relation to the welfare, social, and ecological impacts of such activities on the animals, human participants, people close to them, and the environment [5,8]. In addition, while activities such as horseracing have existed for centuries and are conducted in many countries throughout the world, organizations that govern historic partnerships with horses in sport must embrace the growing impact of public concern for animal welfare. Bergmann [5] argues that interspecies sustainability perspectives that appropriately incorporate animals mean that many current activities, including thoroughbred breeding and racing, are inherently unsustainable and should be discontinued. This perspective was shared before revelations from various jurisdictions that: the fate of many ex-track Thoroughbreds in Australia was unacceptable [9]; the median length of jockeys’ careers was only two years [10]; and that the world endurance racing championships had to be cancelled six weeks before they commenced due to the drought in the region and the likelihood that water being needed for humans would be used to cool and rehydrate horses [11].

Additionally, organizations that govern historic partnerships with horses in sport must embrace the growing importance of public regard for animal welfare. They are under some pressure to change the rules of engagement with horses. The “Enhanced One Welfare Framework” developed in this article reveals significant costs and benefits associated with racing. However, relative to calls to ban horseracing and similar activities as part of sustainability approaches that focus chiefly on animals, the “Enhanced One Welfare Framework” is better positioned politically to guide discussions that renegotiate the conditions under which horses are used for sport and the impact racing has on humans and the planet. This, in itself, is significant, but it is worth noting the scale of the Thoroughbred breeding and racing industry and thereby emphasizing how important these proposed changes are to improving conditions for horses, humans, and the planet. To take just one example, globally in 2019, there were over 140,000 horse races on the flat in 48 countries plus the Hong Kong and Macau Special Administrative Regions and more than 8000 jump races in 16 countries [12].

In this article, we consider the impacts of current Thoroughbred breeding and racing practices in various jurisdictions. We cover practices that are similar to those in other horse breeding and equestrian activities and those that are unique to Thoroughbred breeding and racing. The stakeholders we include are those horses directly affected by these industries, including breeding animals, horses in preparation for racing, racing horses, and, where possible, off-the-track issues for racing horses’ post-racing lives. The article’s scope considers all hands-on personnel: jockeys, track-work riders, trainers, stud personnel, veterinarians, grooms, drivers, and grounds-maintenance staff. The physical environmental issues range in scale from those with immediate impact on the horse (such as straw bedding, manure removal, and so on) to larger-scale issues that impact and are impacted by Thoroughbred breeding and racing, including climate change, horse/personnel transport, energy and waste issues at racing venues, and the issue of “green-washing/sport-washing”.

The following section of this article introduces the One Welfare construct. We critique and enhance this approach by considering improvements that relate to activities involving animals in the sport, leisure, and entertainment industries. We then articulate this Enhanced One Welfare construct with the Five Domains model that was developed primarily for animal welfare assessment. Section 3 of the article summarizes the major welfare impacts relating to the Thoroughbreds themselves (animal welfare). Section 4 then outlines significant welfare issues relating to people (human welfare), while Section 5 presents some of the key environmental issues pertaining to Thoroughbred breeding and racing (environmental welfare), before we conclude with commentary (in Section 6) on the efficacy of articulating the Enhanced One Welfare approach with the Five Domains and identify further important research to be done to enhance the welfare of horses, humans, and the environment.

## 2. Developing One Welfare and the Five Domains

### 2.1. One Welfare

Animal welfare scientists increasingly contextualize their findings within a “One Welfare” framework that enables multidisciplinary collaboration with colleagues in agriculture, business and marketing, the social sciences, medicine, public health, environmental studies, animal studies, and law [13]. As the younger sibling of the One Health construct, One Welfare recognizes that animal welfare, biodiversity, and the environment are intertwined with human wellbeing and community resilience. Sitting alongside and complementing One Health, One Welfare was conceptualized for veterinarians [14] but, more broadly, offers a foundational concept through which to promote the ethical treatment of any animals that are affected by human care. Of course, health, welfare, and well-being are intertwined, but animals can be in good health but nevertheless have poor welfare. So, for there to be a primary focus on welfare, those adopting the “One Welfare” construct may need to assume that stakeholders are of at least average health. That said, the Five Domains construct that we use here in concert with the One Welfare framework requires that health be considered a physical domain that feeds into the overall affective state.

One Welfare has ten outcomes for human, animal, and biodiversity well-being [13]. They are listed as follows: reduction in animal and human abuse; improved animal welfare addressing social problems; links between improved animal welfare and food safety; improved animal welfare—improved human wellbeing; more efficient multidisciplinary approaches; improved life chances—human rehabilitation and animal rehoming; Improved animal and farmer welfare—improved farming productivity; improved animal welfare—addressing poverty and local community support—improving food security and sustainability—and increased biodiversity—improving human wellbeing.

Further, these outcomes formed part of the One Welfare umbrella, which considered companion animals, wildlife, and production/working animals under the image of an umbrella that, at the level, covered animal welfare, human wellbeing, and environment conservation [13]. The absence of animals for sport, leisure, and entertainment within this umbrella means that, similar to the World Commission on Environment and Development, ‘food security’ was prioritised, and ‘working animals in local communities’ implied that the use of ‘working animals’ constituted a low level of economic development, which generally translates into lower environmental impacts in relation to processes such as carbon emissions unless, as highlighted in the model, it occurs through the clearing of forests and the loss of biodiversity. While this focuses attention on biodiversity loss, in industries that are often associated with wealth, such as Thoroughbred breeding and racing, the environmental impacts are more likely to relate to climate, energy, waste, and water.

The One Welfare approach has been updated by García Pinillos [15] to highlight issues in relation to the COVID-19 pandemic. However, the framework is still focused on the humane transport and slaughter of animals, the importance of companion animals in assisting with issues of isolation and depression, and the mixed impacts of the pandemic on biodiversity, with positive impacts due to decreased pollution and reduced impacts of tourism, but increased poaching due to inadequate law enforcement [15]. With a focus on racing, it is possible that the discontinuation of racing during the COVID-19 pandemic may have resulted in numerous welfare issues related to housing, exercise, and adaptation of the musculoskeletal system of horses that were not able to train or that had their training programs disrupted. Additionally, climate change is now included in the One Welfare approach, as it is with other authors [16,17] and organizations (see RSPCA [18]) using this framework. Nevertheless, the absence of recognition of animals in entertainment and sport is critical, especially given that in Australia [19] and parts of the US [20], during even the most severe pandemic lockdowns, the Thoroughbred racing industry was allowed to continue racing, albeit with crowd and travel restrictions.

This article considers the impact of current Thoroughbred racing and breeding practices through a One Welfare lens, as shown in Figure 1. It focuses on practices that are more common in international Thoroughbred breeding and racing than in other forms of horse breeding and equestrian activity. The equine stakeholders we include are those horses directly affected by these industries, including breeding animals, those in preparation for racing, and those racing. The scope embraces all hands-on personnel considered: jockeys, track-work riders, trainers, stud personnel, veterinarians, grooms, drivers, and grounds-maintenance staff. The impacts that current industry practices have on the planet are also considered. The discussion includes an attempt to provide a roadmap for a sustainable future for the racing and breeding industries, albeit one that may currently be unrecognisable. Adopting this model allows discourse around shifts in achievable practice that offer racing and breeding frameworks for a sustainable future.

### 2.2. Five Domains

There are merits to using the Five Domains model of animal welfare assessment [21] as a framework to consider racehorse welfare. Since 1997, in New Zealand, this model has been used to assess the welfare impacts of all research and teaching conducted on vertebrates [22,23]. It has also been used to assess the impact of vertebrate pest control methods [24,25]. There is great potential for this approach to be extended to other domesticated and captive species. For example, it has also been applied to routine equine husbandry and veterinary interventions [26]. The domains of the most up-to-date model are: 1 nutrition, 2 physical environment, 3 health, 4 behavioral interactions, and 5 mental state. The first four domains focus attention on factors that give rise to specific negative or positive subjective experiences (affects), which contribute to the animal’s mental state, as evaluated in domain 5 [27]. Notably, domain 4, now named “behavioral interactions”, focuses on evidence of animals seeking specific goals when interacting behaviorally with (1) the environment, (2) other non-human animals, and (3) humans. It is acknowledged that factors considered within different domains may overlap; for example, an injury may be identified in Domains 2 and 3. 

In 2020, the 57-country International Federation of Horseracing Authorities (IFHA) issued its minimum horse welfare standards [28] and based them on the Five Domains model. This document positions lifelong welfare as “fundamentally important to the viability and sustainability of the industry”. Although the IFHA standards do not have formal status, they are intended to act as a guide for racing authority members.

In addition, most racing Thoroughbreds are fed limited forage (to reduce bulk that would otherwise be carried in the gastrointestinal tract and likely reduce speed), are confined to their stables for 23 h per day [29], have gastric ulcers [30], have negligible direct contact with conspecifics [29], and have limited agency to locomote without human control [31]. Although this does not necessarily apply to all trainers or jurisdictions or under more recent racing and training conditions, these characteristics of racehorse husbandry mean that all four of the physical domains are routinely suboptimal and compromise the mental domain such that, overall, welfare cannot be positive. Nevertheless, while embracing the Five Domains model in its minimum standards, the IFHA affirms the “central role of the horse in racing and accordingly regards the health and welfare of racehorses, in all stages of life, as being fundamentally important to the viability and sustainability of the industry” [28].

Furthermore, articulating the Five Domains with the One Welfare approach that considers the animal, human, and environmental impacts of sport, leisure, and entertainment industries produces an enhanced One Welfare concept that we adopt in this paper. While the domains are applicable for animal and human welfare, as demonstrated below, the wider environmental issues pertaining to sport, leisure, and entertainment industries involving the use of animals need to be considered outside of the domains framework. Importantly, we do not simply adopt the One Welfare approach per se, but instead we critique the model’s historic lack of focus on sport and enhance it by including sport, leisure, and entertainment and by articulating it with the Five Domains model.

## 3. Impacts on Horses

Many participants in the Thoroughbred breeding and racing industries care deeply about horses and develop profound emotional relationships with them, but they must do so in the face of cultures and practices that compromise welfare [32]. Throughout the racehorse’s life, factors that influence welfare relate particularly to husbandry, transportation, handling, veterinary interventions, and injuries.

Premium care of horses in training is what the best trainers are rightly praised for. They know how to condition their horses to align peak performance with key race dates and how to avoid overtraining [33]. They attend to the health needs of the animals in their care because they have no inclination to risk sub-optimal performance that may result from disease or injury. This level of care brings with it employment for many support staff and the considerable costs of training, stabling, feeding, medicating, grooming, and transporting horses for racing. 

Despite premium health care, elite flat racing in some jurisdictions is associated with more fatalities than at lower levels of competition. For example, when racing on turf (as distinct from other surfaces), horses in Group 1 races had 3.19 (95% CI 1.71–5.93) times the odds of fatality compared with horses not racing at this level [34]. In what appears to be a concerning reflection of current practice, there is evidence that horses that were more likely to sustain a fracture when racing were being whipped [35,36] or were reluctant to start [36]. Taken in tandem, these findings seem to underline that, inherent in racing, there is a need to push horses towards their physiological and physical breaking points, and that some of the horses we force to these breaking points are not willing to run. Furthermore, the confinement and the rations fed to horses in training are thought to contribute to the prevalence of gastric ulcers and stereotypic behaviour problems [31].

As indicated earlier, we see the need to consider the impacts on stakeholders of breeding and racing separately.

### 3.1. Breeding—Four Domains

The four physical domains that horses encounter on stud farms (where the emphasis is on ensuring that mares become pregnant and that foals are produced safely and as ready as is appropriate for pre-training) and prior to entering training yards (when incentives lie in producing youngsters that are well-grown and possibly even over-nourished [37]) are quite distinct from those in training. 

#### 3.1.1. Nutrition

In contrast to racehorses in training, broodmares are often provided with access to pasture [31]. This allows them to spend up to 17 h foraging [31]. Stallions, on the other hand, are more likely to be stabled for prolonged periods [31]. This reflects their relative monetary value and the perceived risk of injuries from harmful interactions with infrastructure and other horses when at pasture. Paradoxically, when turnedout to pasture, previously confined horses have an increased risk of explosive displays of exuberance and locomotor injuries because of the likelihood of post-inhibitory rebound [38]. So, the practice of confinement encourages the practice of confinement. However, access to ad libitum grazing and browsing should allow horses to meet their needs for fiber, but local micronutrient concentrations mean that optimal nutrition may rely on appropriate supplementation [39].

#### 3.1.2. Physical Environment

Unlike in the free-ranging state, most domestic stallions lead solitary lives in stables or small paddocks due to concern about injuries and aggressive behavior. In addition, recent evidence shows that stallions can safely live in groups when not surrounded by oestrous mares [40]. Popular Thoroughbred stallions may cover three mares per day, seven days a week, during the breeding season [41]. The best stallions may have no breaks between their naturally day-length dependent breeding seasons because they are flown to the other hemisphere to breed throughout the year [41,42].

#### 3.1.3. Health

About a fortnight after covering, the mare will undergo an ultrasound examination whether she is pregnant, and to determine if she is carrying twins [43]. If she has conceived twins, she is highly likely to lose the pregnancy [44]. Very few twin pregnancies go to term, and those that do often result in foals with poor performance prospects [43]. So, if twins are discovered, the accepted veterinary practice is to destroy one of them by manual compression per rectum [44].

Given that twinning is heritable, with heritability estimates varying from 0.24 to 0.29 [45], this veterinary intervention allows the surviving embryo to go to term, but unfortunately, this also allows the genes for twinning to pass on to the next generation. It appears that Thoroughbreds are especially susceptible to twinning, with approx. 3.5% of pregnancies being twins [46]. Indeed, there is evidence of a possible single gene for twinning in Thoroughbred horses in that the frequency of twinning in half-bred horses increases with the contribution of Thoroughbred genes [46]. Nevertheless, anecdotally, some veterinarians argue that the evidence of increased twinning in Thoroughbred mares, as a consequence of elective veterinary interventions, is lacking. 

The risk of twinning is higher in mares who have already had multiple pregnancies ([47], cited in [46]), while some other evidence relates it to the age of the mare [48]. Other consequences of breeding from an older mare include an inversely correlated risk of her offspring incurring catastrophic fractures when racing [49], an increased risk of neonatal disease [50], and reduced performance if they reach the track [51]. 

Further, not all attempts to breed a racehorse are fruitful. One highly cited historic study indicated that less than half of Thoroughbred foals actually make it to the racecourse [52]. Of those that do, 39.7% retire from racing per year [53]. There is recent evidence that this may, in part, reflect the deleterious effects of inbreeding. A study of the DNA of over 6000 Thoroughbreds born in Europe and Australia found that, for every 10% increase in DNA-measured inbreeding, there was a 48% decrease in the predicted odds of a horse making it to the track [54]. The study also identified a single genetic marker with a significantly negative effect on racing. More than 10% of Thoroughbreds carry the marker, but only 1% of those in the study population were homozygous for it (i.e., carried a ‘double copy’ of it).

It is as yet unclear whether this genetic marker is associated with injury risk, but the marker is close to a gene associated with bone development and repair in other species [55]; hinting that it may relate to the risk of musculoskeletal injury. So, while it may affect individual horse welfare for a small number of horses, it has a considerable impact on the sustainability of the Thoroughbred population overall. However, since Thoroughbreds are now so closely genetically related, the challenge for breeders is to calculate with acceptable accuracy how much risky inbreeding will arise from any given mating. The study’s authors argue that identification of the single genetic marker with a strong deleterious effect on racing means that the unwanted gene can be managed through DNA testing [54]. In essence, if the ‘carrier’ status of both mare and stallion is known, breeders can avoid producing foals with two copies of the genetic marker. However, even the purest of pure-bred animals is likely to be carrying deleterious genes, and the greater the level of inbreeding, the greater the chance of breeding horses with inherited defects [56]. However, avoiding the 10% of horses that are carriers will mean that the gene pool effectively becomes shallower and that, as a result, there is an increased risk of other harmful phenotypes emerging.

#### 3.1.4. Behavioral Interactions 

As noted above, the 2020 model of the Five Domains framework adopts a three-pronged approach to behavioral interactions: with conspecifics (in this case, other horses); other species; and the environment.

##### Behavioral Interactions with Other Horses

At any horse stud, a common aim is to achieve conception on the first covering [44]. This saves time and money for mare owners and allows the stud to take more mares. Most Thoroughbred studs rely on a “teaser” stallion to perform courtship behavior and elicit telltale signs of oestrus in mares [31]. Teasers are usually ponies that are too small to physically mate with the mares [31]. Mares who are not ready to breed may respond aggressively, so it is the teaser’s job to withstand any rejections instead of the stud’s valuable stallions, who may command service fees in the hundreds of thousands of dollars [31]. When we consider animal welfare in the Thoroughbred breeding sector, we should not overlook the welfare of teasers.

##### Behavioral Interactions with Humans

When the mare is ready to be mated, she will be brought to the covering shed. To minimize kicking, she may be fitted with breeding hobbles or boots that limit the movement of her hindlegs. A breeding cape protects her neck from bites during copulation. A device known as a “twitch” may be used as an additional form of restraint. This is a loop of string or rope that is twisted tightly around the upper lip, causing a temporary reduction in heart rate and the release of endorphins that induce calmness in the horse [57]. Some observers regard this level of restraint as a means that justifies the end, but others see its use in diminishing an unreceptive mare’s resistance as tantamount to the rape of the mare [58]. Allowing the mare to reveal when she is so profoundly in oestrus that she will voluntarily stand for mating is the notionally ideal alternative to restraint, but the appetite for risk among stud managers is insufficient to expect improved mare welfare in this regard.

##### Behavioral Interactions with the Environment

There are other potential welfare risks associated with natural rather than artificial breeding methods. For example, some distress may be related to the long-distance transport of breeding stallions, especially the elite, so-called “shuttle”, stallions that cross the equator twice a year to await the return of mares to oestrus in either hemisphere [41]. That being the case, the arguments for introducing artificial insemination into Thoroughbred breeding, as is the case in the breeding of horses for harness racing, may be part of a suite of changes that could enhance the welfare credentials of the Thoroughbred industry.

### 3.2. Racing—Four Domains

Holmes and Brown [32] note that the fate of all equids post-competition varies but generally involves retirement, retraining, breeding, meat/butchery, abandonment or neglect, or euthanasia. The so-called wastage (the term generally used to describe the attrition of Thoroughbreds from industry participation) in the breeding and racing industries can include horses that never race, horses that are injured or killed in racing and training, and horses that are retired from racing every year [59]. 

#### 3.2.1. Nutrition 

The low-forage, high-concentrate diets of racehorses in training are far from what they evolved to consume and all too often lead directly to gastrointestinal disease, some of which can be managed with medication. Across all equestrian pursuits, the highest prevalence of equine squamous gastric disease (ESGD) (that includes gastric ulcers) occurs in Thoroughbred racehorses, with 37% of untrained horses affected, a percentage that increases to 80-100% within 2–3 months of entering race training [30,60,61,62]. Under most racing jurisdictions, oral anti-ulcer medication is largely permitted for racing horses (e.g., [63]). This exemption has been questioned [64], not least because it seems to imply that the priority is on medicating the symptoms rather than addressing the cause, lest doing so detract from race-day performance.

#### 3.2.2. Physical Environment 

It is important to consider aspects of the physical environment that cause negative effects before determining whether they cause disease, and regardless of whether they do cause it. Examples from Mellor et al. [27] that relate to stabling for extended periods, such as confinement of racehorses in training, include close confinement, aversive odors, air pollutants (such as ammonia and dust), physical limits on rest and sleep, and the monotony of the stable’s lighting and physical environment. The average racehorse spends more time in confinement than any other class of horse, with the exception of the breeding stallion and pregnant mares that are sometimes confined long-term for urine harvesting for human pharmaceuticals to produce estrogen and hormone replacement drugs.

#### 3.2.3. Health 

Approximately 10% of racehorses in UK and Australian studies develop stereotypic behaviors [29,65]. Sometimes considered the equine analogue to human obsessive-compulsive disorders, these are repetitive, invariant, and apparently functionless behaviors that reflect historic deficits in welfare and that rarely resolve [66]. 

In order to maintain fitness, racehorses require a high level of veterinary care and farriery. Nevertheless, their workload takes its toll, especially on young horses that are skeletally immature. Approximately 85% of Thoroughbreds suffer from at least one illness or injury during their two- and three-year-old racing seasons [67], with “shin soreness” being the most common problem, affecting 42% of the horses at least once. However, sore or buck skins are the result of stress fractures in the anterior aspect of the cannon bones in the horse’s legs.

Those horses that can withstand early training are more likely to make it to the track as youngsters, and this reality can confound studies that claim to show that early racing is good for the musculoskeletal system [68]. There is a critical causality issue to address here. Being durable enough to begin racing as a two-year old does not demonstrate that racing as a two-year-old enhanced a horse’s durability. In most other equestrian sports, such as those under the Fédération Équestre Internationale (FEI) administration, ponies are the youngest equids allowed to compete, and they are not allowed to compete until they are six years old [11]. Any move to delay the start of racing careers until skeletal maturity would likely be resisted by the breeding sector, whose business models depend on the current levels of turnover.

In addition, race training in only one direction rather than both may contribute to musculoskeletal wear-and-tear [69]. For example, it may lead to the greater loading of the leading front leg in asymmetric gaits, notably the canter [70]. That said, we cannot rule out innate asymmetries that have been reported more in Thoroughbreds than other breeds [71]. A small study of NZ Thoroughbreds [72] has shown mild asymmetry, with more horses having a flat left foot (10 of the 75) than a right foot (4 of the 75). The prevalence of left-to-right asymmetry in this pilot study was 18%, which was higher than the 8% reported in a population of Dutch Warmblood horses thought to show natural laterality associated with preferred stance [73].

Consequently, the very notion of racing to the horses’ physiological and physical limits brings with it the risk of injury. The reported incidence of horse fatalities during Thoroughbred flat racing is up to 2.44 per 1000 starts [74], and the majority of fatalities have been attributed to musculoskeletal injuries necessitating euthanasia [75].

As we will note later in this article, the risks of injuries and fatalities among horses are closely linked to the risk for jockeys [76]. Here we note that the paradox for jockeys is that, in contrast to the horses, they are aware of the risks and that, despite their awareness, they are also required to whip horses that are in a competitive position in a race and may fall as a consequence and endanger the jockeys themselves.

#### 3.2.4. Behavioral Interactions 

As indicated earlier, the 4th domain was formerly labeled the “behavior domain”, but since 2020 it has been subdivided into three distinct categories to separate interactions with other horses, with humans, and with the immediate environment.

##### Behavioral Interactions with Other Horses

While in training, racehorses’ interactions with other horses are generally reduced to an absolute minimum. This is in a bid to reduce injuries during agonistic interactions and reflects the monetary value of racehorses, the scale of training fees, and the likelihood of any injury compromising performance [31].

##### Behavioral Interactions with Humans

The interface between humans and racehorses involves behavior-modifying equipment used in thoroughbred racing. A study of the behavior and apparatus that horses wear when racing revealed associations among certain items of gear and horse performance on the day [77]. Without claiming causality, it showed poor performance was associated with the use of boots, bandages, jaw-encircling nose bands, nose rolls, and pacifiers. It also showed that horses that were difficult to handle were likely to underperform.

The potential for gear to affect performance is fundamental to the integrity of racing. The rules of racing state that permission to use any piece of approved gear other than basic snaffle bits has to be given prior to the race by the stewards [63]. That said, the scientific community has only recently begun to put ancient and modern theories on horse handling and training to the test in a bid to identify which techniques and devices work and why [78,79,80]. A good example of a questionable item of gear is the so-called “tail chain”, which is a short length of metal chain secured to the top of the horse’s tail by a rubber band and then hangs between the horse’s buttocks. This equipment has not been studied empirically. Anecdotally, it is believed to dissuade the horse from taking air into its rectum as it gallops, thereby preventing putative abdominal pain and associated poor performance. There is no peer-reviewed evidence of horses achieving this feat. However, given the anatomy of the horse’s gastro-intestinal tract, it seems unlikely that such air intake could affect performance in this way, or alternatively that a tail chain could reduce any such effect. It is possible that the chain hitting the soft tissues of the perineal area may motivate the horse to gallop harder, which could be seen as performance-enhancing [81]. 

The items or equipment that have received particular attention from equine welfare scholars since the emergence of equitation science [82] include whips, tongue ties, and bits. Only Norway prohibits the carrying of whips in all flat races involving horses three years of age and over [83]. In contrast, 98% of horses in Australia will be whipped when racing [84], especially as they slow down at the end of the race. Given that integrity and safety are critical, it is hardly surprising that whip use has been normalized in horse racing. That said, there is no reliable peer-reviewed evidence that whip use makes racing safer [85]. 

The whipping of animals for public entertainment and financial gain, some of which entails problem gambling, is problematic from moral standpoints [83,86]. Ethically, there are members of the public who perceive the whipping of racehorses to be cruel [87,88]. Their concerns can be contextualized within broader socio-historical trends in the education of animals and humans away from corporal punishment [89] and towards positive reinforcement [90]. Indeed, the use of an aversive stimulus, such as whipping, as “encouragement”, may be more akin to models of corporal punishment of humans now largely abandoned as they are considered outdated, unethical, and ineffective [89].

The tongue-ties are elastic, nylon or leather bands, that are wrapped around the tongue of a horse, fixing it to the mandible [91]. Historically, they have been recommended for safety because they stop the tongue from traveling over the bit, but they are primarily applied as a conservative treatment for upper airway obstruction, particularly dorsal displacement of the soft palate [92]. This condition involves the free caudal margin of the soft palate becoming dislodged from its normal sub-epiglottic position during exercise, resulting in an obstruction to airflow. It can lead to impaired athletic performance due to reduced oxygen supply to the exercising muscles [91]. The dorsal displacement of the soft palate is one of the most common forms of dynamic upper airway collapse affecting racehorses [93] and is estimated to affect up to 20% of racehorses [94]. Importantly, the exact prevalence of the condition is elusive because a definitive diagnosis cannot be made without an endoscopic examination during exercise.

The potential physical and psychological harm that tongue-ties (in terms of compromised vascular perfusion and restricted deglutition) can inflict on horses is of concern from a welfare perspective. Tongue-ties have been shown to have a significant negative effect on a horse’s physiological and behavioral state [95]. Concerns about the effect of tongue-ties on equine welfare have led to their use being banned in equestrian disciplines, by the FEI since 2004 and, for racing, in Germany since 2018 [96]. More recently, Racing Australia [63] abolished the use of nylon stocking tongue-ties in Thoroughbred racing, although other materials are still allowed. Nylon was believed to be especially likely to cut into the surface of the tongue and could be more readily overtightened. Tongue-ties of various designs are also still widely used in racing elsewhere in the world, such as in the UK, where reportedly 5% of horses race with a tongue-tie [97]. This is in contrast to Australia, where 20% of the population has tongue-ties [98]. Such a variable international uptake of the use of this device internationally reflects the lack of science underpinning its use.

The industry needs to address two separate aspects of current tongue-tie use. There are many causes of breathing noise that are unrelated to palatal issues, which would not be helped by a tongue-tie [98]. Secondly, there is the issue of control. If one argues that tongue-ties are needed for safety because they stop the tongue from travelling over the bit, then theoretically one is obliged to use them for *all* horses, since all have the capacity to adopt this evasion. The evasion of the bit is resented by many horse riders, but gaping, gagging, and tongue-lolling are normal responses among horses that have yet to habituate to bit pressure, and one that overlooks the aversiveness of having metal structures sit on soft oral tissues even before rein tension is applied. Nevertheless, every racehorse must wear a bit in their mouth. More than 60 different designs of bit are permitted in racing [63]. The main purpose of a bit is to apply discomfort to the tongue and lower jaw of the horse to motivate it to change its speed or direction. Many of the bits on the approved list are simple, traditional designs, whereas others are complex pieces of engineering with flanges, clips, and jaw-encircling structures. These are intended to address specific behavioral problems such as lugging (veering to one side) or over-galloping (galloping with a high head position while straining at the bit). 

The horse is an obligate nasal breather, and optimal airway is thought to depend on the lips being sealed, a seal that is disrupted by the bit [99]. Most naïve horses react to the bit with manifestations of discomfort or pain [99], so there is an argument for allowing horses to race without a bit. The rise of bitless riding in non-racing pursuits offers a template for racing codes that permit horses to race without a bit in their mouth.

##### Interactions with the Physical Environment

The racehorses’ interactions with the environment are largely managed by humans. The resultant safety issues may include falls and injuries associated with, for example, hitting the running rail and conflicting behaviors in the starting gates [100]. That said, even beyond racing, traveling at high speeds in close proximity to unfamiliar horses will always bring with it an increased risk of collisions with infrastructure compared with other equestrian sports, with the possible exceptions of jumping events, polo, and polo-crosse.

## 4. Human Welfare

The benefits of racing for humans include employment, prize money, tourism income, promotion of a town’s name and identity, taxes to the state and federal governments, annual revenue to the country’s economy, and the opportunity to attend social gatherings. Millions of people enjoy the spectacle of race days and the thrill of watching equine athletes compete [42]. Additionally, many people thrive in racing yards and are highly motivated to work with horses. However, jockey and track-work rider falls and deaths are of great concern, as are some of these workers’ employment conditions. 

Asking horse enthusiasts to whip equids as part of their work creates internal conflict, especially if it contributes to the risk of a high-speed fall [36]. Other costs to strappers (grooms) include long work hours, horse-related injuries, and poor air quality, issues that have not been addressed extensively in published literature. For workers in a yard who develop strong attachments to specific horses, the turnover of horses in yards may have an impact on mental health, but this has not been explored as there are few who have studied the role of human-equine attachment. Other issues that have not been reported in published literature are the impact of gambling and gambling addiction, loss of assets, and family breakdowns that may occur when jockeys have retired. 

The health and well-being of the human workers engaged in the racing industry has been studied primarily in relation to the jockeys. The few studies that included track workers and workers on Thoroughbred farms have focused primarily on injuries and falls [101,102,103,104], and, again using a Five Domains approach, these will be discussed below.

### 4.1. Nutrition

Weight requirements for professional jockeys require maintenance of specific daily weight guidelines [105]. In order to reach their goal weights, jockeys report using several chronic and acute methods, including sporadic eating, calorie restriction, purging, fluid restriction, the use of diuretics and laxatives, and regular use of sweatsuits and saunas [105,106,107]. The concerns regarding the consequences of these eating patterns reflect a wide range of potential health issues that include physical and mental health [105,106].

The “relative energy deficiency in sport” (RED-S) syndrome is a condition where male athletes may have impaired physiological function related to low energy availability and may impact jockeys due to the weight requirements that result in periodic food deprivation [107]. Male flat jockeys have been reported to have low bone mineral density (BMD) by several researchers [105,108,109]. However, resting metabolic rate has not been reported to differ from that predicted in jockeys [107]. Therefore, resting metabolic rate and hip and spine BMD in apprentice and senior jockeys were compared to test whether low energy availability and a consistent lack of osteogenic stimulus due to years of riding (which Wilson et al. [107] regarded as a non-weight bearing activity) resulted in RED-S and provided an explanation for the low BMD findings [107]. The authors found no evidence to support those claims that symptoms of RED-S accounted for previously reported low BMD in jockeys, as the resting energy measurements were within normal range for both apprentice and senior jockeys. They suggested that the poor bone health reported is due to a lack of osteogenic stimulation rather than low energy availability, but suggested the hypothesis needs further testing [107].

The so-called backstretch area of the racetracks is where horses are stabled and cared for. In the United States, backstretch workers have reported issues with diet and nutrition as well [101]. The access to fruits and vegetables among backstretch workers (grooms, hot walkers) living on site was limited, and while workers reported preferring to prepare their own food, lack of electricity and refrigeration resulted in workers cooking on charcoal grills and table stoves [101]. In addition, low-cost options available to workers included fast-food restaurants, which contributed to reports of unhealthy diets [101]. Health issues related to nutrition included overweight concerns, diabetes, and cardiovascular disease [101].

### 4.2. Work Environment

Crowded conditions for horses put workers at an increased risk of kicks and bites. When work surfaces are wet, human workers can be at risk for slips, trips, and falls; the lighting conditions in stables can increase these risks as well. Organic dust and inorganic dust create hazards that may result in acute or chronic respiratory diseases. Extreme temperatures can cause problems for workers as well, whether it is heat or cold. Despite the importance of these issues, there is sparse literature assessing them specifically in the context of Thoroughbred farms and racing settings.

The injuries related to race days have been assessed more often than off-track incidents. McCrory et al. [110] compared fatal and nonfatal injuries among jump and flat professional jockeys in Great Britain and France and concluded that although French jockeys experienced fewer falls, they were more likely to sustain an injury because of the falls. The authors attributed the differences in risk of injuries to circumstances related to the track surfaces and to the greater number of horses in flat races in France. In the most comprehensive review published for the United States, between 1993-1996, there were a total of 606 injury events per 1000 jockey-years that occurred during official Thoroughbred races [111].

The physical locations on tracks where jockey injuries occur during racing have also been assessed. Ryan et al. [112] reported race day injuries among jockeys in Maryland and found that 41% of injuries occurred in the vicinity of the starting gate, with the homestretch, post finish line, and finish line being other locations where injuries commonly occurred. Waller et al. [111] reported that the most frequent location where injuries occurred was entering, within, or leaving the starting gate.

Other aspects of the physical environment associated with welfare risks for workers in the industry overlap with the risks to horses, including those encountered in stables and outdoors. Among backstretch workers at racetracks, workers reported respiratory problems and eye irritations attributed to dust particles from sawdust and wood shavings [101]. These have not been reported in relation to jockey health and safety but include increased potential for respiratory diseases and heat stress. The track surfaces, especially synthetic tracks, vary in composition, but all contain some combination of synthetic fibers that may represent inhalation hazards for riders [113].

### 4.3. Health

The bone health and body composition associated with jockey practices of weight restriction were reviewed [114]. Jockeys have significant challenges related to weight making during racing seasons, as well as needing to have proper weight for the horses they ride. This has been documented to impact their physical health through fat composition, bone health, and cardiovascular disease. Dunne et al. [114] concluded that the fat mass of jockeys has increased in recent decades and may lead to less healthy weight loss approaches in the future. In addition, male flat race jockeys have lower bone density than national hunt and female jockeys [114]. Most importantly, the authors recommend the development of international standard protocols for assessment of body composition and bone health to accurately assess bone status [114].

Mental health problems have also been discussed, with a focus on mood, anxiety, distress, disordered eating, and substance misuse [115,116,117]. Professional jockeys reported greater psychological distress, depression, anxiety, and perceived stress compared to amateur jockeys [116]. The factors associated with higher depression scores included current injuries, social anxiety, and high perceived stress scores [116]. In addition, jockeys reported higher levels of depressive and anxiety symptoms and alcohol abuse compared to other elite athletes [115]. Other factors associated with mental health difficulties among jockeys included career dissatisfaction and contemplating retirement [115]. Further, lower competitive riding weights were also associated with more disordered eating attitudes and negative impacts on mood [115]. Of concern was the fact that help-seeking for mental health disorders was low among jockeys [115].

In a survey among backstretch workers, primary health concerns included musculoskeletal pain, gastrointestinal illnesses, hypertension, diabetes, vision problems, and respiratory diseases [101]. The workers also reported dental/oral health issues and a lack of access to dental services [101]. 

In 1989, the Ryan Family Foundation supported on-site assistance programs at Thoroughbred racetracks focused on the needs of horse care workers impacted by alcohol and drug use [118]. The documentation of the extent of these problems has received limited attention, with only a handful of published reports among this vulnerable population of workers [101,118,119]. This is an area that should be addressed to ensure the sustainability of the industry by supporting the workers involved.

### 4.4. Behavioural Interactions

Here we consider the interactions industry personnel have with other humans, horses, and the work environment [113,120,121,122,123,124].

#### 4.4.1. Behavioral Interactions between Industry Personnel and Other Humans

The nature of the racing industry involves some severe issues related to contractual agreements, financial security, medical care provision, and retirement benefits for jockeys and low-wage earners in the industry. Jockeys are independent contractors who have agents who negotiate with trainers and owners for permission to ride their horses [121]. The payment structure depends on base fees and the potential for a percentage of winnings, which makes the professional jockey similar to a piecemeal wage earner. Further, novice jockeys may work long hours for minimal pay. Other workers in the industry may also be subject to low wages [101,122].

#### 4.4.2. Behavioral Interactions between Industry Personnel and Horses

Although many adverse health outcomes have been suggested as potential issues among jockeys and other workers actively engaged with horses in racing [113], falls and traumatic injuries have received a significant amount of attention in published literature [76,102,103,104,112,115,125,126,127,128,129]. The types of injuries that have been reported include soft tissue, concussions, and lower limb injuries, with concussions being the most serious [76,102,123,126,127,128,129]. Further, fractures and head injuries being those most associated with the end of the jockeys’ careers [125]. The race-related injuries were reported to result in more costly claims than non-race claims, and those resulting in head injuries were the costliest [126] and the deadliest [127]. Comparing fatal and nonfatal injuries among flat jockeys in France and Great Britain, McCrory et al. [110] reported fractures, dislocations, and concussions were higher in France and suggested one reason for the differences seen was the number of horses in the field.

In the Southeastern United States, Swanberg et al. [103] studied Latino and non-Latino workers on Thoroughbred farms and reported horse-related injuries were common and most often involved injuries to the head and chest. Filby et al. [104] studied accidents to staff in racing stables in Great Britain and reported associations between staff size and number of horses and accidents. In Tasmania, track-work riders were reported to be at increased risk of injury compared to jockeys [102].

A number of factors that increase the risk of injury might be addressed to reduce injuries, including reaction time, hand grip strength, riding style, aerobic and anaerobic fitness, height and weight, and total body fat [102]. Legg et al. [130] recommended developing metrics to address needed standards to enhance performance based on rider strength and to improve conditioning programs for jockeys.

As discussed in the welfare section related to horses, there have been studies related to the use of whips and other aggressive approaches to training racehorses. While there has been a significant amount written about the role of attachment between animals and people, the impact of requiring the use of aggression in racing on people has not been addressed in the literature. Nor has there been anything published on the grief of the jockeys, trainers, or handlers when a horse must be euthanized. As one of the issues that has been raised in sustaining racing as an industry due to the concerns of the public and the widely publicized deaths of racehorses during prominent races, this is an area of work that should be further developed. Grief and loss can have long-term impacts on the health and well-being of all involved.

#### 4.4.3. Behavioral Interactions between Industry Personnel and the Work Environment

Chronic stress can impact reaction time and accurate decision making [121]. Jockeys have been reported to have high perceived stress, high injuries related to their work, and exhausting work schedules. Landolt et al. [121] used the effort-reward imbalance (ERI) work stress model to assess the impact of low and high stress periods on decision-making, physiological measurements of stress (cortisol and alpha amylase), and the cardiovascular stress response. ERIS was associated with decision-making in the high stress period but not in the low-stress period [121]. However, the decreased perception of rewards (status, esteem, and money) was associated with decreases in decision-making in both the high and low stress periods [121]. Further work is needed to connect the impact of findings such as these to actual work performance, such as injuries and other adverse health outcomes.

## 5. Impacts on the Environment

The Thoroughbred industry, similar to wider equestrian cultures, both impacts and is impacted by the environment due to its reliance on natural resources, its vulnerability to major weather and climate events, and the effects of the industry on various components of the environment [42,131]. While the application of technology and the adoption of new practices modify these relationships, environmental factors remain significant. Water, power, fertilizers, and pesticides used in the maintenance of monocultures of grass on stud farms and racetracks may have an impact on the environment. Many human commercial activities negatively impact the environment, so it is important to acknowledge that racing and breeding are not unusual in this regard. Horse transport requires the use of fossil fuels, and the Thoroughbred breeding industry is unique in its requirement for “natural breeding”, which sees stallions shipped across the globe twice per year with significant environmental impact relative to comparable activities.

Further, as Miller, [132] p. 2 observed, “sports leave their own ecological mark and provide symbolic cover for more significant polluters”. Given the myriad of environmental relationships in which the Thoroughbred breeding and racing industries are engaged, the lack of recent academic literature is most surprising. Other sports that use fossil fuels or draw large crowds to particular venues have been the subject of considerable academic inquiry [132,133]. Some of the environmental issues present in these sports overlap those in Thoroughbred breeding and racing.

The British Horseracing Authority (BHA) engaged consultants White Griffin to prepare “an initial assessment of the sport’s progress on environmental sustainability, to help support and inform the industry’s long-term planning” [134]. There was a period of wider public engagement in early 2022, with the report released in June 2022 [135]. The White Griffin report is the latest example of engagement with environmental and sustainability issues, with a number of other bodies in different locations having made moves to engage with such issues over the past decade or longer. For example, the Victoria Racing Club (in Australia) has a sustainability charter that dates to 2007 [136]. The Hong Kong Jockey Club includes their direct impacts on issues such as water, waste management (from the 1400 horses in their stables), energy, procurement, and carbon emissions in their environmental report, and importantly also notes that “in addition to its own initiatives, the Club makes a significant environmental contribution through its Charities Trust” [137] p. 3. 

White Griffin [135] classified the major environmental sustainability issues in British Thoroughbred racing as fossil fuels and greenhouse gas emissions, water availability and extreme weather, biodiversity and land use, waste and recycling, commercial partnerships and the supply chain, and reputation management and social responsibility. Such an extensive list is important in that it shows recognition of waste, water, and energy issues at racetracks while acknowledging larger-scale issues, environmental impacts beyond the immediate racing environment, and the more intangible impacts, such as the sector’s reputation and social license to operate. The following sub-headings vary slightly from those used by White Griffin [135] but capture many of the same issues.

### 5.1. Anthropogenic Climate Change

One of the most pressing issues that is likely to impact the entire Thoroughbred industry, albeit in various ways, is anthropogenic climate change. British Horseracing Authority Chief Regulatory Officer Brant Dunshea observed in early 2022 that “climate change and sustainability were major challenges for our sport, given how dependent it was on the environment, transportation, and the use of essential resources” [138,139]. As noted by Thompson et al. [131], “projected increases in average temperature and duration of heatwaves… are particularly concerning”. This is because heat stress in hot racing environments such as parts of Australia, some US states, the Middle East, and elsewhere is a significant concern for horse welfare in addition to impacting performance and behavior. Humidity can compound heat stress in horses [131], making locations such as Hong Kong, Singapore, and north-east Australia, among others, particularly susceptible to this issue. Hot weather policy and guidelines have existed for over a decade in Thoroughbred racing [131,140]. Various cooling strategies to mitigate this (ranging from changing the season and time of day/evening when races are held through to the use of water for cooling) are plausible, but in the absence of concerted action limiting climate change, such adaptation approaches will become essential. While this focuses attention on biodiversity loss, the intensive nature of contemporary animal protein production and, to a lesser extent, leisure industries such as Thoroughbred breeding and racing, in richer countries tend to generate environmental impacts related to climate, energy, waste, and water.

It is not only on the racetrack in front of crowds and cameras that horses will suffer from extreme heat conditions, but the movement of horses to race (and for breeding purposes, given the stallion and mares have to be physically together) is likely to contribute to heat stress. Thoroughbreds may be enclosed in trailers for extended periods with limited air circulation, thereby posing a risk to horse welfare. This is an issue not only in the countries that are historically associated with hot conditions. In July 2009, the first abandonment of a race meeting in the United Kingdom due to heat occurred at Worcester, “with the track running out of water, one horse collapsing, and other horses arriving at the track dehydrated” [42] p. 127. The transport of Thoroughbreds for breeding and racing is not only affected by the environment but also has deleterious impacts beyond those already discussed. International transportation of Thoroughbreds by air, in particular for breeding and racing contributes to the racing, industry’s carbon footprint [131].

Climate change may impact Thoroughbred breeding in the development of the horse and the breeding industry’s ability to avoid wastage that arises from breeding too many suboptimal foals. Research in Japan found differences in growth rates in colts and fillies between hotter and cooler climates [141]. So, while climate change may actually be favorable in some locations, the increased heat may have detrimental impacts in other locations that are currently favorable for rearing Thoroughbreds.

### 5.2. Water

The use of water by the Thoroughbred breeding and racing industry is of growing importance as the planet faces the consequences of anthropogenic climate change. Racetracks, auction venues, and breeding establishments all use significant amounts of water [135,142]. Sometimes this is arguably justifiable in softening hard (“fast”) racing tracks to limit the potential of injuries to Thoroughbreds. On other occasions, while there may be some arguments on food and welfare grounds, a lot of the watering of paddocks and gardens is for aesthetic purposes. While particular grasses are grown for food at horse studs, Thoroughbreds are often given supplementary nutrition at farms and particularly in racing stables [31,142]. The Thoroughbred breeding and racing industry projects an image of care and sophistication at racetracks, auction venues, and stud farm landscapes with a view to attracting potential participants and retaining those already engaged with the industry, and this image often relies on the use of water [142,143].

Thoroughbred breeders are responding to climate change and water issues in various ways. In Australia, one leading stud farm, Arrowfield, gained increased access to existing water supplies by constructing a pipeline to obtain water from Glenbawn Dam as part of a “drought-proofing” strategy [42,142]. Major Thoroughbred breeders, such as Darley Australia, have established Thoroughbred farms in different states, in part to limit “the impacts of seasonal extremities and long-term climate change by being able to adjust stocking rates across all their properties or shift horses if required” [42] p. 126. Water issues, in particular, will impact the costs of Thoroughbred breeding and potentially affect profitability, as water issues are increasingly related to climate change. If welfare concerns such as the “full circle” or “whole of life” approach to the breeding, ownership, and death of Thoroughbreds are taken seriously [144], then “off the track” rehousing programs for “retired” horses will also likely be impacted by the above considerations.

### 5.3. Energy

Energy use in horse racing is often associated primarily with lighting, heating, cooling, communications, and so on at racetracks. This is important, but the use of energy to produce food for horses, to transport horses (including international shuttle stallions that are flown between hemispheres), and for breeding farm and racing yard maintenance is more significant. There is the additional energy use relating to crowds attending race venues (some of which is possible by public transport but often involves private automobile use) and the travel for work purposes of jockeys and track personnel. White Griffin [135] p. 3 claims that in Britain, “the average jockey clocks up more than 40,000 miles annually and is almost entirely reliant on road travel”. Some initiatives introduced by various horse racing authorities to reduce energy use will be identified later in this article.

### 5.4. Waste

White Griffin [135] p. 12 observed that “most people who participated in this study cited waste and recycling as their foremost environmental priorities. Nonbiodegradable single-use plastics—cups, bottles, plates, carrier bags, etc.—are perhaps the most prominent examples”. Waste, or litter, is a visible and genuine environmental problem, but the fossil-fuel input into single-use plastics is often overlooked. Other forms of waste include the limited life of plastic running rails at racetracks or the hurdles used in jump racing, but they are perhaps less obvious [135]. The horses also “produce on average 50 pounds of manure each day” [135] p. 12, which has environmental and health impacts, particularly in concentrated environments such as stables as opposed to open fields. In this context, more research is needed on the environmental and human health impacts of veterinary drugs used in the Thoroughbred breeding and racing industries, given that this is a problem associated with intensive agricultural industries such as poultry and pig farming. There are also opportunities for addressing waste issues beyond the actual industry. For example, the Charities Trust funds projects such as place-based environmental education in Hong Kong, infrastructure upgrades at a nature reserve, and static and mobile recycling units that extend beyond the actual facilities owned and operated by the Hong Kong Jockey Club (see [42,137]).

### 5.5. Greenwashing

A range of initiatives are being undertaken by various organizations in different jurisdictions and segments of the Thoroughbred industry, but the central challenge for the Thoroughbred breeding and racing industry will be to undertake genuine sustainability actions and avoid accusations of “greenwashing”, i.e., where a corporation portrays itself as more environmentally minded than its actions warrant. Miller [132] argues that various professional sports have significant carbon footprints through stadium construction, energy use, and player and spectator travel. Further, he posits that sports associations have indirect, but significant, impacts through their sponsorship deals with “the supremely craven gas and petroleum industries”, which provide corporations with a positive image [132] p. 2. The British Horseracing Authority and other Thoroughbred industry bodies are moving towards having environmental, social, and governance (ESG) processes in place, but to ensure a genuine environmental welfare approach, they will have to engage with the most significant environmental issues and ensure that there are no inconsistencies in their own procurement and sponsorship practices that could be seen to foster environmental harm.

## 6. Sustaining the Industry

In this section, we propose innovations that may help Thoroughbred breeding and racing assure a sustainable future. One critical aspect of any discourse on sustainability is identifying what is to be sustained. The answer could include the planet, particular species, individual members of a species, corporate profits, employment, and even a way of life. In this article, we are highlighting that change is needed and that the only way these industries will survive is if they reform and align with (changing) community expectations related to animal welfare, human welfare, and environmental welfare.

### 6.1. Celebrating Racehorse Durability and Planning Ahead for a Second Career

Thoroughbred horses have been bred over centuries for speed and stamina. This allows them to do one thing better than most other members of their species: run. That said, it has recently been shown that, just as they can be bred for speed, Thoroughbreds can also be bred for durability, the ability to withstand the rigours of training and racing [145]. Logically, the longer horses are in racing and the more starts they have, the more likely they are to risk injury. That said, if the market comes to value durability as much as other performance traits, the sector can reward breeders who select for long racing careers alongside other attributes.

Australia is a leading producer of two-year-old racehorses, and there are rich rewards for the breeders of the next star of the track [42]. Valuing durability requires a shift from the current emphasis on finding the latest and greatest young horse each year for events such as the Golden Slipper (for two-year-olds), the Gold Coast Magic Millions (for two-year-olds), and The Oaks (for three-year-old fillies). With such an emphasis on races for young horses, it is unsurprising that one of the most recent audits of Australian Thoroughbreds revealed that only 7.74% had racing careers that lasted longer than 4 years [67]. It has been suggested that if the industry were to put the major prizes in place chiefly for the fastest eight-, nine-, or ten-year-old Thoroughbreds, one would anticipate a dramatic drop in wastage [146]. Trainers would have an incentive to celebrate their most durable horses and redouble efforts to avoid the career-threatening injuries that remove many young Thoroughbreds from racing early in their careers. Breeders would be rewarded for breeding champions that win as veterans. Furthermore, albeit with a reduced size of the annual foal crop and the global racing herd. Clearly, this gesture towards long-term sustainability could be met with resistance from the breeding sector, which relies on turnover to sustain demand. As it happens, the Melbourne Cup is a case in point. The parade of past winners and associated activities present an opportunity to see some perennial stars of the turf, some of whom have competed in multiple Melbourne Cup races over the years.

Ideally, the athleticism, sensitivity, and versatility of Thoroughbreds make them ideal horses for a variety of equestrian disciplines, for both pleasure and professional riders. However, most Thoroughbreds begin their training with a singular focus on racing, and there are significant differences between the behaviors that make for a successful racehorse and those suitable for recreational riding [147]. For example, due to the unusual position of the riders’ legs, the bit pressure cues (i.e., via the reins) delivered to control a racehorse differ widely from standard practices among recreational riders. Racehorses are often ridden with strong tension through the reins, which, when released, becomes a signal to accelerate. In contrast, acceleration cues in recreational riding are given by a rider’s legs. One of the side effects of using strong rein tension is that horses learn to habituate to, and may ultimately ignore, rein cues unless extremely strong pressures are applied. In a recreational context, this can make them unsafe to ride if they take fright, because a leisure rider may not be able to pull hard enough on the reins to get the horse to slow down. In another example, race jockeys usually mount their horse while it walks around. The short stirrups found on racing gear do not allow mounting from the ground, and Thoroughbreds are unused to the feel of a rider’s leg against their side [148]. Former racehorses have to be retrained to accept the pressure of a rider’s legs resting on the lateral thoracic wall in the non-racing position. 

Additionally, even after retraining, the behavioral legacies of their racing career can make some racehorses unsuitable for inexperienced or recreational riders, limiting their post-racing career options. Other problem behaviors that can arise as legacies of a racing career include difficulty turning in circles, head-tossing, rearing, bucking, and overexcitement at shows or events. In response, in many countries, there are increasing off-the-track-Thoroughbred (OTT) rehoming initiatives, funded by the racing industries. The success of these programs depends, not least, on there being sufficient good, knowledgeable homes for horses. This means that the riding market now has to somehow absorb the horses that were not bred primarily for equitation. This is important because there is evidence that the leisure horse market regards OTT Thoroughbreds differently from other potential riding horses [147,148]. 

In Australia, for the 2017-18 season, 11,177 Thoroughbreds were registered, which would have led to an estimated 5000 geldings needing a new home outside the racing industry [146]. These horses need to be comprehensively retrained if they are to become safe riding horses [149]. This can take at least four weeks and cost upwards of $1000 for medical care alone prior to retraining [150]. As a result, to underwrite these horses’ transition to a second career, the industry would need to provide at least $5 million per year for retraining the retired geldings alone, if they were all viable for non-racing equestrian careers [151].

The suggestions above are just some of the options that could help reduce wastage in the industry and provide a better life for horses during and after racing. Others include Australia’s proposed National Horse Traceability Register [152], which that would track a horse’s journey throughout its life. This would reveal the ultimate fate of Thoroughbreds—specifically, how many are actually suitable as equestrian or pony club mounts, and how many ultimately have no other value than as meat or non-ridden companions? As such, these data would have merit in guiding the calculation of an ethical cap on the annual foal crop of all breeds.

As we have become familiar with the reality of driverless cars, it should not surprise us to learn that remote-controlled mechanical jockeys are now commonplace in camel racing, having replaced boys (who historically were enlisted when they were as young as four years old) in the United Arab Emirates and Qatar since 2005 [153]. Operated remotely by a trainer traveling trackside in a motor vehicle, these mechanisms include heart rate monitors and a whipping function, but only a single rein (on the left side) for steering and deceleration. It is not apparent whether there are any deleterious consequences for camel welfare. Such a radical undertaking in Thoroughbred racing would garner dismay from those who value horsemanship but consider jockey welfare to be worth risking. Clearly, it would require a complete redesign of the race-tracks to accommodate the number of motor vehicles that would be needed based on the number of horses in a race. Furthermore, while it may well be that this re-development was much easier for camel race-tracks than it would be for established tracks for horses, it would add to emissions. That said, it is possible that the use of drones that allow trainers to observe and steer their horses may represent a future development that preserves human life.

### 6.2. Measuring and Monitoring SLO That Embraces One Welfare

Modern grass-roots participatory democracy has triggered conversations around the concept of ‘social license to operate’ (SLO) [149]. This concept has existed in resource-extraction industries for a number of years but is not without controversy, given its potential to focus on local interests and exclude more remote and indirect stakeholders from the conversation. Its application to human–animal relationships, and particularly the use of animals in sport, is relatively new [154]. Nevertheless, the racing industry is paying attention [134]. Conversations around SLO in the horses-in-sport domain must renegotiate the conditions under which humans use horses in sporting activities and acknowledge the risks to horses as conscripts to sport, to amateur riders as volunteers, and to jockeys and trackwork riders as professionals. As the most televised form of horse sport in most developed countries, racing must earn the trust of all stakeholders, including the public. Doing so demands transparency of operations, the establishment and communication of shared values, and a demonstration of competence [155,156]. It also requires leadership in contentious areas where trust has been undermined. Catastrophic falls and the use of the whip are the issues foremost in the minds of the public.

Furthermore, discussing SLO can be uncomfortable for industry participants, so it cannot take place in a vacuum of leadership, but the difficulties are multiplied if no genuine conversation takes place. As digital literacy improves, more voices will contribute to the world-wide conversation about the complexities and opportunities of people partnering with horses in sport. Such discourse must not dismiss the voices from outside the industry as merely uneducated. A worthy goal should be to agree on what measures of SLO are valid and how they can be monitored, since, in an ever-changing world, we need metrics that demonstrate approval. Such metrics, at least for racing, may need to reflect some of the costs and benefits we have highlighted in this article. Additionally, organizations overseeing racing may take responsibility for improving horse and rider welfare through their everyday practices. They could undertake to report annual SLO metrics along with data on incremental improvements in one welfare-related outcome such as the prevalence of falls, ESGD, and stereotypic behaviors.

Some racing organizations outside the Thoroughbred sector are exploring new initiatives. For example, Harness Racing Australia [157] has adopted a combination of leadership, consultation, and the leveraging of institutionalized administrative structures to overcome barriers to SLO discourse. Such consultations must agree on what measures of SLO are indicative of approval and on the conditions that affect the mental domain of Thoroughbreds from cradle to grave and can be monitored (and externally audited, as required). Committing to reporting on and incrementally improving one welfare metric may be good for everyone.

### 6.3. Industry Initiatives on Horse Welfare

In 2020, the Horse Welfare Board (HWB, established in 2019) at the request of the BHA produced a five-year plan focused on the vision that “Respect for the horse is at the heart of everything [they] do: every horse bred for racing will enjoy a life well lived” [134]. It reported that public polling suggested that fatalities and the whip were the areas of greatest public concern, while a “lack of information and engagement has encouraged the growth of negative perceptions”. The report committed to measuring horses’ quality of life and wellbeing; increasing the use of data to prevent injury and manage safety; and establishing lifetime responsibility for Thoroughbreds, based on a plan to trace every racehorse from birth (presumably till death). The board responsible for the report is positioned as being independent of the BHA.

There have been similar calls for action from within the Thoroughbred sector itself, arguably more so from the breeding industry than from the racing industry. For example, in Australia, in response to The Final Race, an Australian Broadcasting Corporation [9] exposé about the fate of ex-track Thoroughbreds, Thoroughbred Breeders Australia (TBA) convened a ground-breaking project, the Thoroughbred Welfare Initiative (TWI), specifically directed at addressing the aftercare challenges facing the Thoroughbred breeding and racing industries. Its report, titled “The most important participant: A Framework for Thoroughbred Welfare” [144], offered 46 recommendations as practical ways to deliver improved whole-of-life welfare outcomes and ensure the welfare of every Thoroughbred horse is protected from birth until death. Seven actions were proposed to span these 46 individual recommendations as the signature reforms of this report (see Box 1).

Box 1Actions to deliver improved whole-of-life welfare outcomes for Thoroughbreds in Australia as proposed by the Thoroughbred Welfare Initiative (2021).*Responsibility:* The Thoroughbred Industry should accept responsibility and take all reasonable steps to ensure its horses have a good life, including after racing, and a humane death.*National Standards*: Governments should develop, with the support of industry, national welfare standards for all horses. While other species such as cattle and sheep have enforceable standards, these do not exist for horses. This would mandate minimum care for horses at all stages of life, including for Thoroughbreds after they exit racing and breeding.*Industry Standards*: The industry should develop enforceable national welfare standards for all Thoroughbred horses while racing and breeding. These standards would be appropriate to all stages of life (i.e., breeding, pre-training, racing, etc.). These would make clear to all participants the minimum care they are required to provide to remain in the industry.*Traceability:* Governments should create a national traceability register for all horses that identifies each horse individually as well as its location and owner. Without such a register, it is almost impossible to have an effective welfare regime. It is also important for the Thoroughbred industry to know where its horses are in retirement.*Transition:* The Thoroughbred industry should make further investments in programs to transition horses out of racing and breeding. Moving Thoroughbreds into good homes or a purposeful second career is important for welfare. While there are lots of good programs assisting this transition, the industry needs to ensure it is doing all it can to stimulate demand for Thoroughbreds.*Safety Net:* The industry should establish a national Thoroughbred safety net to support horses at risk of poor welfare outcomes after leaving racing and breeding. Such a safety net would allow the industry to help those horses that it no longer has any jurisdiction over.*National Body:* The industry should create a national welfare body—with the proposed name of Thoroughbred Welfare Australia—to drive improved outcomes for Thoroughbreds at all stages of their lives.

### 6.4. Opportunities for Assuring Horse Welfare

High-welfare racing may attract sponsors and punters, just as humanely raised meat attracts consumers. One possible feature of high-welfare races would be rules that permit jockeys to carry whips but to use them only for safety purposes. Whip-free races have been shown to be as fast and safe as those in which whipping for “encouragement” is allowed [85]. As it happens, for decades, the BHA has run a hands-and-heels series of races for apprentice jockeys that set an excellent precedent for high welfare racing. The conditions for hands-and-heels races are the same as for standard whip races, with the following exceptions: the horses must be suitable for apprentices, and whips can be held but must not be used, unless required to get a reluctant horse moving at the start of a race or for safety reasons. If the whip is used for any reason, it is subject to an inquiry from the stewards. The hands-and-heels races were created to foster best practice in race-riding without a reliance on the whip. They also provide a unique opportunity to analyse the relationship between steering/safety, and whip use being permitted. Thompson and colleagues used these races for a comparative study, whereas a rigorous randomized control trial would lack validity and be unable to be blinded. After matching races against seven criteria (including racecourse, distance, number of horses, race class, and track condition), Thompson et al. [85] compared stewards’ reports from 67 hands-and-heels races with 59 reports from case-matched races where whipping was permitted. They found there were no statistically significant differences for movement on course (a proxy for quality of steering), interference between horses, jockey-related incidents, or race times. That is, there was no evidence that whip use improves steering, reduces interference, increases safety, or improves finishing times. These findings suggest that whipping-free races do not compromise racing integrity. By the same token, bit-free racing merits trialing, at the very least.

Racing of horses maintained on pasture, rather than in stables, may represent a means of testing Thoroughbred performance while still providing horses with the friends, freedom, and forage that World Horse Welfare, an international charity that provides welfare advice to the BHA and FEI, sees as the horses’ basic needs. However, given the general agreement that foraging reduces speed, this prospect seems unlikely in commercial racing. A proxy for good nutrition may be the prevalence of ESGD. 

### 6.5. Opportunities for Assuring Human Welfare

Several authors have proposed approaches to improving the health and well-being of jockeys engaged in Thoroughbred racing [107,123]. Recommendations included improved strength and aerobic training to increase optimal performance and address the mechanical and physical demands of race riding. While there are concussion protocols in place in some jurisdictions, Ryan et al. [123] proposed the development and use of a “Return to Ride” protocol, much as the one developed for concussed athletes, to determine when athletes can safely return to their sport, which is lacking in the United States. Addressing the issue of weight restrictions is more complicated, and some have proposed increasing jockey minimum weights [123], while others have proposed developing diets that specifically address the jockey weight management issues in racing to decrease disordered eating patterns [107]. Acknowledging that jockeys do not have access to the same quality of medical care that is standard for similarly profitable sports organizations is a first step to addressing this issue and may reduce health challenges among this group of elite athletes.

The 1948 Universal Declaration of Human Rights contains several articles related to human rights associated with working conditions [158]. Article 22 addresses social security and covers economic, social, and cultural rights; Article 23 expands this to include just and favorable conditions of work and remuneration and the right to form trade unions; and Article 25 establishes the right to a standard of living adequate for oneself and one’s family (https://www.un.org/en/about-us/universal-declaration-of-human-rights, accessed on 27 January 2023). There are no universal guidelines established for jockeys to protect their interests that cover all jockeys throughout the world. The wages for workers in the racing industry are not known, and finding the average annual wages for jockeys is not easily accessible in the United States [123]. Although there are standards and minimum wages in some jurisdictions [120,124], it is difficult to work out the annual income of a jockey because of the seasonal nature of racing. In addition, retirement benefits are likely to vary by country, but it is unclear how these differences impact the welfare of the working population. Disabling injuries are highly likely to occur among jockeys, but it is not clear how they are compensated and supported after a catastrophic, career ending injury. As described above, there are many work-related issues that result in adverse health impacts, that are both physical and emotional that should be addressed to support more favorable outcomes for those working in racing. Simply applying principles from the Universal Declaration of Human Rights to the population of workers in Thoroughbred breeding and racing would begin to address the welfare of these workers.

### 6.6. Initiatives by Horseracing Authorities to Address Environmental Issues

One example of environment and sustainability gaining prominence in the thought processes and actions of Thoroughbred industry participants is the recent work by The Jockey Club in the United Kingdom. The Jockey Club, founded in 1750, has responsibilities for racing, training, and breeding in the United Kingdom, with 15 racecourses (including both flat racing and jumps racing venues), training grounds at Newmarket, Lambourn, and Epsom, and The National Stud, which stands stallions. In 2012, the Jockey Club launched its “Going Green” program, which included targets for energy and waste [159,160]. The concept behind Going Green was to make “environmental management everyone’s responsibility” [159]. To this end, The Jockey Club supports “over 42 active and engaged Green Champions” using a “detailed and interactive toolkit [that] was developed by Carbon Intelligence with guidance, tools, and resources” [160]. 

The target for waste was to send no waste to landfills by 2020, which was achieved in 2018. Five racecourses in the United Kingdom operated by The Jockey Club achieved more than 80% recycling in 2018, with the best achieving 92%. The target for 2019—to reduce the total volume of waste by 10%—was exceeded [159]. The target for energy was to reduce energy consumption by 25%, but the wider intent was to not only “reduce the energy we use but ensure that the energy we do use comes from clean and sustainable sources” [160]. 

## 7. Conclusions

Our world is changing, and so too is our relationship with horses and our regard for their welfare. Organizations that govern human-horse relationships, especially when horses are used for entertainment, will have to move with and drive these changes or face the prospect of finding themselves driven by external forces. Adopting an enhanced One Welfare lens extends this conceptual tool to be applicable to the sports and entertainment industries, including the Thoroughbred breeding and racing industries. As we have demonstrated, these industries differ in their treatment of animals, the construction of preferred environments for these activities, and the impacts on people, the planet, and profits, compared with agricultural activities and other environments where “working animals” are located. In addition, the use of this concept enables us to identify key animal, human, and environmental issues and offer potential ways to address these important issues. Doing so is politically more likely to succeed than calls to immediately ban Thoroughbred breeding and racing industries everywhere, which would also have significant social and commercial impacts. The adoption of our suggested approach will enable ongoing research into the effects of the major changes that are implemented and research into areas where further knowledge is needed prior to changes being made. Given the importance of One Welfare for the well-being of horses, humans, and the environment, the existence of incomplete knowledge or the absence of consensus should not hinder the adoption of important changes that need to be made. 

## Figures and Tables

**Figure 1 animals-13-00490-f001:**
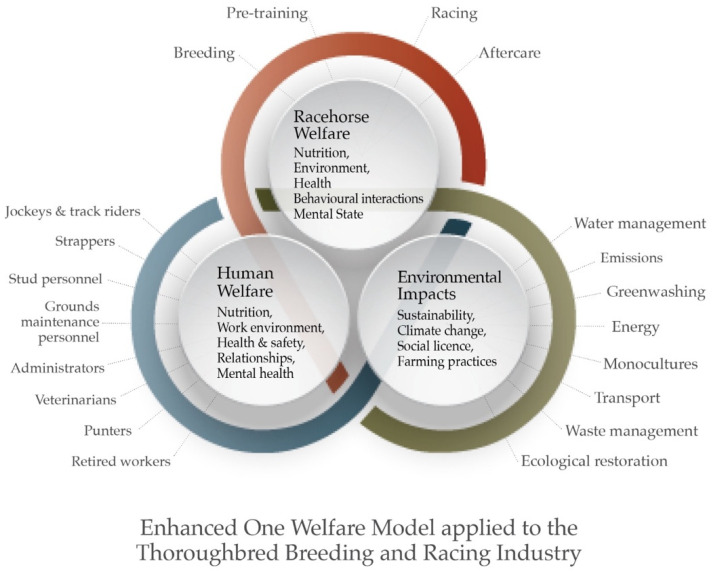
An Enhanced One Welfare Model applied to the Thoroughbred breeding and racing industry. Source: The Authors in conjunction with Cristina Luz Wilkins, Horses and People, who drew the figure.

## Data Availability

Not applicable.

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
