# Peer review of "Sustainability and the Thoroughbred Breeding and Racing Industries: An Enhanced One Welfare Perspective"

_animals, 2023, doi:10.3390/ani13030490_

Round 1
Reviewer 1 Report
This is a very interesting summary of the current situation of Thoroughbred industry based on the One Welfare perspective.
P4L158: Figure 1 (instead of one).
P4: Figure 1 should be before 2.1 Five Domains text (L167-168) instead of after 2.1.
As a suggestion, same concepts of Figure 1 colud be used in the text (Physical environment instead of Environment; P12L515: Human Welfare instead of Impacts on Humans; P13L567: Work environment instead of physical environment; P13L598: Health and Safety…and so on).
P6L217-219: maybe include training as a factor that influence horse welfare.
P6L240: 3.1. Breeding – Four Domains
P6-7: no comments regarding overweight or obesity is considered for breeding. It would be interesting to include it.
P8L317 Behavioural interaction with other horses: some comments about stallions and mares could be included.
P8L354: some comments regarding stereotypic behaviour related to nutrition (physical environment and health) could be included, considering reports in racehorses. A brief comment if on L376-379, but interesting studies could be mentioned to show how management in racehorses impact on this behaviour.
P9L391: Fédération Équestre Internationale (FEI)…and FEI onwards.
P9L392: equids are allowed to compete, and they are not allowed…
P9L402: in this pilot study was 18% higher than the 8%...
P11L475: Tongue-ties have been…
P11L478: FEI
P11L479: since 2018 (97). (dot missing) More recently…
P11L487-489: even though some studies have shown some effect of tongue-tie over DDSP, they are not conclusive…therefore, some comments about that could be included in the discussion.
As a suggestion, Four domains discussion about breeding and racing could be merged….some aspects are similar, and the ones that are not, could be compared. Problems in both Thoroughbred live stages could be better reflected (and compared), making the discussion more robust. Also, the text of breeding and racing is a little long, and sometimes hard to follow…merging both would enhance the discussion.
P12L556: They suggested that the reported poor bone health…
P14L646: (77, 103, 104…). (dot missing)
P14L646-650: types of injuries are repeated in those lines…also, those reports could maybe be included in 4.3. Health and Safety?
P15L684: not instead of nor?
P15L701: Miller is between parentheses.
P17L766: as the planet faces the consequences of anthropogenic…
P18L824: The BHA
P18L855: nine- or ten-year old Thoroughbred
P18L859: win as veterans. Albeit….
P20L962: BHA produced a five-year…
P21L980: Six actions were proposed (table 1 shows 7 actions).
P22L1017: As it happens, for decades, the BHA…
P22L1040-1042: some very interesting new studies have shown the possibility to maintain racehorses on pasture- or high-fiber diets…these studies could be included to present this may be a considerable option to give these horses a more natural type diet instead of high-energy ones.
P22L1049-1050: some redaction is needed in that sentence.
P23L1067: States (124) (bracket missing)
P23L1073-1074: health impacts, that are both physical and emotional, that should be addressed…
P23L1093-1095: some comments on the results of that initiative would be appreciated.
Author Response
Reviewer #1
This is a very interesting summary of the current situation of Thoroughbred industry based on the One Welfare perspective.
P4L158: Figure 1 (instead of one). Edited in the revised manuscript.
P4: Figure 1 should be before 2.1 Five Domains text (L167-168) instead of after 2.1.
As a suggestion, same concepts of Figure 1 colud be used in the text (Physical environment instead of Environment; P12L515: Human Welfare instead of Impacts on Humans; P13L567: Work environment instead of physical environment; P13L598: Health and Safety…and so on). Thank you for this excellent suggestion. Changes have been made in the revised manuscript.
P6L217-219: maybe include training as a factor that influence horse welfare.
Thank you for this excellent suggestion. The sentence in question now reads as follows: Throughout the racehorse’s life, factors that influence welfare relate particularly to husbandry, transportation, handling and veterinary interventions, medications, training and injuries.
P6L240: 3.1. Breeding – Four Domains Edited to uppercase B in the revised manuscript.
P6-7: no comments regarding overweight or obesity is considered for breeding. It would be interesting to include it. Thank you for this interesting suggestion. In the absence of reliable body condition scores and bodyweights for breeding mares we are reluctant to weigh-in on this topic.
P8L317 Behavioural interaction with other horses: some comments about stallions and mares could be included. Thank you for this excellent suggestion. We have added a new sentence to address this point. It reads as follows: By the same token, the imposed separation of mares and stallions in most contexts on studs is likely to create frustration and thus compromise welfare.
P8L354: some comments regarding stereotypic behaviour related to nutrition (physical environment and health) could be included, considering reports in racehorses. A brief comment if on L376-379, but interesting studies could be mentioned to show how management in racehorses impact on this behaviour.
We have addressed this with extant copy as follows:
Approximately 10% of racehorses in UK and Australian studies develop stereotypic behaviours [29, 66]. Sometimes considered the equine analogue to human obsessive-compulsive disorders, these are repetitive, invariant and apparently functionless behaviours that reflect historic deficits in welfare and that rarely resolve [67].
P9L391: Fédération Équestre Internationale (FEI)…and FEI onwards. This was changed as suggested in the revised manuscript.
P9L392: equids are allowed to compete, and they are not allowed…Edited as suggested.
P9L402: in this pilot study was 18% higher than the 8%... Thank you for this excellent suggestion. The sentence in question now reads as follows: The prevalence of left to right asymmetry in this pilot study was 18%, in contrast to the 8% reported in a population of Dutch Warmblood horses thought to show natural laterality associated with preferred stance [74].
P11L475: Tongue-ties have been…Edited as suggested.
P11L478: FEI Edited as suggested.
P11L479: since 2018 (97). (dot missing) More recently… Edited as suggested.
P11L487-489: even though some studies have shown some effect of tongue-tie over DDSP, they are not conclusive…therefore, some comments about that could be included in the discussion. Thank you for this excellent suggestion. We have added a new sentence to address this point. It reads as follows: This contributes to the problem of studying DDSP clinically and may explain why some research endeavours shown some effect of tongue-tie over DDSP but are not conclusive [93].
As a suggestion, Four domains discussion about breeding and racing could be merged….some aspects are similar, and the ones that are not, could be compared. Problems in both Thoroughbred live stages could be better reflected (and compared), making the discussion more robust. Also, the text of breeding and racing is a little long, and sometimes hard to follow…merging both would enhance the discussion. Thank you for this excellent suggestion. We are content that the headings and sub-headings make logical sense and acknowledge that the length of the breeding and racing sessions reflects the burgeoning literature on the equine stakeholders.
P12L556: They suggested that the reported poor bone health…Edited as suggested.
P14L646: (77, 103, 104…). (dot missing) Edited as suggested.
P14L646-650: types of injuries are repeated in those lines…also, those reports could maybe be included in 4.3. Health and Safety? Thanks for noticing the redundancy. It has been corrected in the paper. Although these could be included in Health and Safety, we wanted to make the point that injuries are an interaction between the human and the horse so chose to place them in the current section.
The types of injuries that have been reported include soft tissue, concussions, and lower limb injuries, with concussion injuries being the most serious [77,103,124,127,128,129,130]. Fractures and head injuries being those most associated with the end of the jockeys' careers [126].
P15L684: not instead of nor? Edited as suggested
P15L701: Miller is between parentheses. Edited as suggested
P17L766: as the planet faces the consequences of anthropogenic… Edited as suggested
P18L824: The BHA Edited as suggested
P18L855: nine- or ten-year old Thoroughbred Edited as suggested
P18L859: win as veterans. Albeit….EDITED AS SUGGESTED
P20L962: BHA produced a five-year…EDITED AS SUGGESTED
P21L980: Six actions were proposed (table 1 shows 7 actions). EDITED AS SUGGESTED
P22L1017: As it happens, for decades, the BHA…EDITED AS SUGGESTED
P22L1040-1042: some very interesting new studies have shown the possibility to maintain racehorses on pasture- or high-fiber diets…these studies could be included to present this may be a considerable option to give these horses a more natural type diet instead of high-energy ones.
Interesting point. We have addressed this with extant copy as follows:
Racing of horses maintained at pasture, rather than in stables, may represent a means of testing Thoroughbred performance while still providing horses with the friends, freedom and forage that World Horse Welfare, an international charity that provides welfare advice to the BHA and FEI, sees as the basic needs of all horses. However, given the general agreement that forage reduces speed, this prospect seems unlikely in commercial racing. A proxy for good nutrition may be the prevalence of ESGD.
P22L1049-1050: some redaction is needed in that sentence.
We have edited the offending sentence as follows:
While there are concussion protocols in place in some jurisdictions, Ryan et al. [124] proposed the development and use of a “Return to Ride” protocol, much like the one developed for concussed athletes to determine when athletes can safely return to a sport.
P23L1067: States (124) (bracket missing) EDITED AS SUGGESTED
P23L1073-1074: health impacts, that are both physical and emotional, that should be addressed…EDITED AS SUGGESTED
P23L1093-1095: some comments on the results of that initiative would be appreciated.
We edited this as follow:
unique opportunity to analyse the relationship between steering/safety and whip use being permitted. Thompson and colleagues used these races for a comparative study, whereas a rigorous randomized control trial would lack validity and would be unable to be blinded. After matching races against seven criteria (including racecourse; distance; number of horses race class and track condition), Thompson et al . [86] compared stewards’ reports from 67 Hands-and-Heels races with 59 reports from case-matched races where whipping was permitted.
Reviewer 2 Report
Overall this is an excellent written review of many of the societal issues and pressures facing the Thoroughbred racing industry. It is an interesting adaptation of One Health and is a very holistic review of an entire industry.
I would suggest that the paper provides an outline at the beginning so that the reader knows that suggested practices which are achievable will be included. The overall negativity it tone as the reader progresses needs to be balanced with some encouragement. Otherwise I fear that the very stakeholders who may need to consider such thoughts will be turned off prior to reading the conclusions.
Line 48 – While not the focus of this paper, might it be important to also mention the use of animals as companions? Certainly horses will fall within this category as well, and the positive impact on human health has been well documented. * At this point in time the authors had not yet addressed companion animals, so appears as a lack of consideration. It may be helpful to insert earlier.
Line 116 – I would specify that animals can be in good visible physical health but have poor welfare – it is hard to imagine a scenario that poor welfare would truly not have any impact on health parameters – perhaps state that there are less visible challenges to health that are not apparent to the observer.
This article frames the impact of the TB industry on climate. While not unimportant, I would put this in context of industries with much larger impacts, such as animal protein production.
Line 139 – Can these thoughts be broken up, That’s a really long sentence.
Line 193 – Similar conditions could be stated to be true of the majority of competition horses
Line 256 – The reality’s of climate and growing conditions also prevent yearly browsing/grazing to meet fiber requirements
Line 364 – Would not prohibiting medication also impact welfare? Especially as horses not in training and on optimal management programs can still develop ulcers?
Line 372 – Other elite competition horses are similarly housed
Line 472 – extra to in sentence
492 – suggest different word choice then resent – perhaps disliked or is interpreted negatively
Bit usage – if discomfort or tongue rolling is associated with an unadapted horse, does this truly imply poor welfare if the horse becomes acclimatized to it? A simple analogy is placing a coat on a dog that freezes momentarily due to the unfamiliar feel, but then after adaptation continues normal behavior. It is the duration of the distress that it is important.
Line 632 – related to contractual agreements
Line 660 – delete it
Line 738 – delete on
Line 741 – missing punctuation relative to references
Line 757 – extra period
Line 766 – add of
Line 847 – it may be helpful to put the location of these events, and to include some of the major events in other countries
Line 881 – delete one
Line 910 – I would suggest deleting the section on robotic jockeys. If the intention of the paper is to encourage change, adding something that would be considered abhorrent by many of the readers may discourage the undertaking of any of the suggested changes. One could compare to the desire to transition to self-driving cars, there are still significant challenges which can increase risk of safety, rather than a net benefit. What is the impact on the camels?
Line 1067 – missing parenthetical bracket
I like the evidence of accomplished change at the end, which leaves the paper on a bit of a higher note. I highly recommend trying to incorporate some positivity throughout the paper as stated earlier. I fear that it may read as a direct attack on an entire industry which may limit reader's acceptability. Present the challenges that exist, but temper with the hope that positive change can yield high impact.
Author Response
Reviewer #2
Overall this is an excellent written review of many of the societal issues and pressures facing the Thoroughbred racing industry. It is an interesting adaptation of One Health and is a very holistic review of an entire industry.
I would suggest that the paper provides an outline at the beginning so that the reader knows that suggested practices which are achievable will be included. The overall negativity it tone as the reader progresses needs to be balanced with some encouragement. Otherwise I fear that the very stakeholders who may need to consider such thoughts will be turned off prior to reading the conclusions.
Thank you for this excellent suggestion. We have added a new sentence to address this point. It reads as follows: Adopting this model allows discourse around shifts in achievable practice that offer racing and breeding frameworks for a sustainable future.
Line 48 – While not the focus of this paper, might it be important to also mention the use of animals as companions? Certainly horses will fall within this category as well, and the positive impact on human health has been well documented. * At this point in time the authors had not yet addressed companion animals, so appears as a lack of consideration. It may be helpful to insert earlier.
Although this is an interesting point, Thoroughbreds engaged in racing are not treated as companion animals. The comment in the manuscript related to the lack of information about how human-animal interactions impact the welfare of the humans but from a different perspective than that used in studying companion animals and humans.
Line 116 – I would specify that animals can be in good visible physical health but have poor welfare – it is hard to imagine a scenario that poor welfare would truly not have any impact on health parameters – perhaps state that there are less visible challenges to health that are not apparent to the observer.
Good point. We have changed this sentence as follows: Of course, health, welfare and well-being are entwined but animals can appear in good health but nevertheless have poor welfare.
This article frames the impact of the TB industry on climate. While not unimportant, I would put this in context of industries with much larger impacts, such as animal protein production.
We added this based on the reviewer’s comments:
While this focuses attention on biodiversity loss, the intensive nature of contemporary animal protein production and to a lesser extent leisure industries, such as Thoroughbred breeding and racing, in richer countries tend to generate environmental impacts related to climate, energy, waste and water.
Line 139 – Can these thoughts be broken up. That’s a really long sentence.
EDITED AS SUGGESTED
Line 193 – Similar conditions could be stated to be true of the majority of competition horses
Fair point. We believe we have effectively covered this point by inference with the extant copy: It [the article] focusses on practices that are more common to international Thoroughbred breeding and racing than to other forms of horse-breeding and equestrian activity.
Line 256 – The reality’s of climate and growing conditions also prevent yearly browsing/grazing to meet fiber requirements
Good point. We have changed this sentence as follows: Where climate permits, access to ad libitum grazing and browsing should allow horses to meet their needs for fibre but local micro-nutrient concentrations mean that optimal nutrition may rely on appropriate supplementation [40].
Line 364 – Would not prohibiting medication also impact welfare? Especially as horses not in training and on optimal management programs can still develop ulcers?
Good point. We have changed this sentence as follows: This exemption has been questioned [65] not least because, even if medication does not itself compromise welfare, it seems to imply that the priority is on medicating the symptoms rather than addressing the cause, lest doing so detracts from race-day performance.
Line 372 – Other elite competition horses are similarly housed
Fair point. As above, we believe we have effectively covered this point by inference with the extant copy: It [the article] focusses on practices that are more common to international Thoroughbred breeding and racing than to other forms of horse-breeding and equestrian activity.
Line 472 – extra to in sentence
EDITED AS SUGGESTED
492 – suggest different word choice then resent – perhaps disliked or is interpreted negatively
Good point. We have changed this sentence as follows: The evasion of the bit is disliked by many horse-riders but gaping, gagging, and tongue-lolling are normal responses among horses that have yet to habituate to bit pressure and one that overlooks the aversiveness of having metal structures sitting on soft oral tissues, even before rein tension is applied.
Bit usage – if discomfort or tongue rolling is associated with an unadapted horse, does this truly imply poor welfare if the horse becomes acclimatized to it? A simple analogy is placing a coat on a dog that freezes momentarily due to the unfamiliar feel, but then after adaptation continues normal behavior. It is the duration of the distress that it is important.
Good point. We have changed this sentence as follows: The horse is an obligate nasal breather and optimal airway patency is thought to depend on the lips being sealed; a seal that is disrupted by the bit [100]. So, there is a physiological argument for allowing horses to race without a bit. And while most naïve horses react to the bit with manifestations of discomfort [100], the eventual disappearance of this behavioural response (as a result of habituation) does not offset the physiological disruption caused by the bit(s). Furthermore, the rise of bitless riding in non-racing pursuits offers a template for racing codes to permit horse to race without a bit in their mouth.
Line 632 – related to contractual agreements EDITED AS SUGGESTED
Line 660 – delete it
EDITED AS SUGGESTED
Line 738 – delete on. EDITED AS SUGGESTED
Line 741 – missing punctuation relative to references EDITED AS SUGGESTED
Line 757 – extra period EDITED AS SUGGESTED
Line 766 – add of EDITED AS SUGGESTED
Line 847 – it may be helpful to put the location of these events, and to include some of the major events in other countries
This was specific for the United Kingdom so the suggestion would not add anything to the article and we decided to keep the text as originally presented to maintain focus on the issues.
Line 881 – delete one
EDITED AS SUGGESTED
Line 910 – I would suggest deleting the section on robotic jockeys. If the intention of the paper is to encourage change, adding something that would be considered abhorrent by many of the readers may discourage the undertaking of any of the suggested changes. One could compare to the desire to transition to self-driving cars, there are still significant challenges which can increase risk of safety, rather than a net benefit. What is the impact on the camels?
Your comments raise some very interesting points. We have changed this para to reflect them, as follows: As we have become familiar with the reality of driverless cars, it should not surprise us to learn that remote-controlled mechanical jockeys are now commonplace in camel racing, having replaced boys (who historically were enlisted when they were as young as four years old) in the United Arab Emirates and Qatar since 2005 [155]. Operated remotely by a trainer travelling trackside in a motor vehicle, these mechanisms include heart rate monitors and a whipping function but only a single rein (on the left side) for steering and deceleration. It is not apparent whether there are any deleterious consequences for camel welfare. Such a radical undertaking in Thoroughbred racing would garner dismay from those who value horsemanship but consider jockey welfare to be worth risking. Clearly, it would require a complete re-design of the race-tracks to accommodate the number of motor vehicles that would be needed based on the number of horses in a race. Furthermore, while it may well be that this re-development was much easier for camel race-tracks than it would be for established tracks for horses it would add to emissions. That said, it is possible that the use of drones that allow trainers to observe and steer their horses may represent a future development that preserves human life.
Line 1067 – missing parenthetical bracket EDITED AS SUGGESTED
I like the evidence of accomplished change at the end, which leaves the paper on a bit of a higher note. I highly recommend trying to incorporate some positivity throughout the paper as stated earlier. I fear that it may read as a direct attack on an entire industry which may limit reader's acceptability. Present the challenges that exist, but temper with the hope that positive change can yield high impact.
Good point. We have changed this sentence as follows: Adopting this model allows discourse around shifts in achievable practice that offer racing and breeding frameworks for a sustainable future.
Reviewer 3 Report
This paper is a commentary on an important topic for the Thoroughbred breeding and racing industries. I appreciate a lot the deep analysis developed by the authors applying the Five Domains Model in the One Welfare Framework.
I notice a little mistake at line 244 (need to close an open bracket).
I would like to invite the authors to consider in their paper the following suggestions:
a) in the section "Impact on horses" it could be useful to consider the negative impact of early training not only from the point of view of the physical development of the horse but also from the point of view of his/her psychological development and attachment to humans. Many behavioral problems could be the result of bad training, and unrespectful of the ethological times of the horse.
In the same section, I think we need also to consider another issue that can impact horse welfare: the misuse of drugs both in breeding and racing.
b) in the session "impacts on humans" authors considered only workers in the field of horse racing. I think it would be useful to add some thoughts about the impact of this field on the public, especially referring to the business linked to gambling and the social problem of gambling addiction.
Lastly, when speaking about the impact on the environment, I suggest the authors consider also the negative effects of the residual of medical products coming from this industry (e.g. antimicrobial resistance and environmental resistome). This issue is maybe less important than in livestock production but it is useful to think about that to improve the environmental sustainability of this sector too.
Author Response
Reviewer #3
This paper is a commentary on an important topic for the Thoroughbred breeding and racing industries. I appreciate a lot the deep analysis developed by the authors applying the Five Domains Model in the One Welfare Framework.
I notice a little mistake at line 244 (need to close an open bracket). EDITED AS SUGGESTED
I would like to invite the authors to consider in their paper the following suggestions:
- a) in the section "Impact on horses" it could be useful to consider the negative impact of early training not only from the point of view of the physical development of the horse but also from the point of view of his/her psychological development and attachment to humans. Many behavioral problems could be the result of bad training, and unrespectful of the ethological times of the horse.
Good point. We have changed this sentence as follows: Throughout the racehorse’s life, factors that influence welfare relate particularly to husbandry, transportation, handling and veterinary interventions, medications, training and injuries.
In the same section, I think we need also to consider another issue that can impact horse welfare: the misuse of drugs both in breeding and racing.
Good point. Please see emended sentence above
- b) in the session "impacts on humans" authors considered only workers in the field of horse racing. I think it would be useful to add some thoughts about the impact of this field on the public, especially referring to the business linked to gambling and the social problem of gambling addiction.
The focus we chose in relation to humans was to address the workers who had direct interactions with the Thoroughbreds. While the issue of gambling is important, it is a different focus than we chose for this manuscript.
Lastly, when speaking about the impact on the environment, I suggest the authors consider also the negative effects of the residual of medical products coming from this industry (e.g. antimicrobial resistance and environmental resistome). This issue is maybe less important than in livestock production but it is useful to think about that to improve the environmental sustainability of this sector too.
We felt this went beyond the scope of the commentary so leave this discussion to others to address this important topic.
Reviewer 4 Report
Some small edits/suggestions:
3.1 Breeding (to be capitalized for consistency)
P9L402,403 was was
P11 L475 to to
P15 L660, 661 it at
P21, L971 972 than than
(there are likely some gambling addiction references in the social science/psychology/social work type of literature)
Author Response
Reviewer #4
Some small edits/suggestions:
3.1 Breeding (to be capitalized for consistency) EDITED AS SUGGESTED
P9L402,403 was was EDITED AS SUGGESTED
P11 L475 to to EDITED AS SUGGESTED
P15 L660, 661 it at EDITED AS SUGGESTED
P21, L971 972 than than. EDITED AS SUGGESTED
(there are likely some gambling addiction references in the social science/psychology/social work type of literature)
The focus we chose in relation to humans was to address the workers who had direct interactions with the Thoroughbreds. While the issue of gambling is important (and we have mentioned it obliquely), it is a different focus than we chose for this manuscript.